# 2022 Overview of Metabolic Epilepsies

**DOI:** 10.3390/genes13030508

**Published:** 2022-03-12

**Authors:** Birute Tumiene, Carlos R. Ferreira, Clara D. M. van Karnebeek

**Affiliations:** 1Faculty of Medicine, Institute of Biomedical Sciences, Vilnius University, LT-03101 Vilnius, Lithuania; 2National Human Genome Research Institute, National Institutes of Health, Bethesda, MD 20892, USA; carlos.ferreira@nih.gov; 3Departments of Pediatrics and Human Genetics, Amsterdam University Medical Centers, University of Amsterdam, 1012 WX Amsterdam, The Netherlands; c.d.vankarnebeek@amsterdamumc.nl; 4Emma Center for Personalized Medicine, 1012 WX Amsterdam, The Netherlands

**Keywords:** inherited metabolic diseases, specific treatments, diagnostics, International Classification of Inherited Metabolic Disorders, congenital disorders of autophagy, disorders of metabolite repair or proofreading, disorders of the synaptic vesicle cycle

## Abstract

Understanding the genetic architecture of metabolic epilepsies is of paramount importance, both to current clinical practice and for the identification of further research directions. The main goals of our study were to identify the scope of metabolic epilepsies and to investigate their clinical presentation, diagnostic approaches and treatments. The International Classification of Inherited Metabolic Disorders and IEMbase were used as a basis for the identification and classification of metabolic epilepsies. Six hundred metabolic epilepsies have been identified, accounting for as much as 37% of all currently described inherited metabolic diseases (IMD). Epilepsy is a particularly common symptom in disorders of energy metabolism, congenital disorders of glycosylation, neurotransmitter disorders, disorders of the synaptic vesicle cycle and some other IMDs. Seizures in metabolic epilepsies may present variably, and most of these disorders are complex and multisystem. Abnormalities in routine laboratory tests and/or metabolic testing may be identified in 70% of all metabolic epilepsies, but in many cases they are non-specific. In total, 111 metabolic epilepsies (18% of all) have specific treatments that may significantly change health outcomes if diagnosed in time. Although metabolic epilepsies comprise an important and significant group of disorders, their real scope and frequency may have been underestimated.

## 1. Introduction

Understanding the genetic architecture of metabolic epilepsies is of paramount importance, both to current clinical practice and for the identification of further research directions. Metabolic epilepsy, defined as epilepsy that results directly from an inherited metabolic disorder (IMD) in which seizures are a core symptom of the disorder [1], are at the intersection of the disciplines of biochemical and molecular genetics and epileptology. In the last decade, high-throughput gene-sequencing studies have yielded an abundance of discoveries that change the paradigms in both epilepsy genetics [2] and IMDs [3]. The International Classification of Inherited Metabolic Disorders (ICIMD), which was recently adopted and endorsed by the international metabolic community, expands the definition of IMDs according to the current understanding of molecular and cell biology and encompasses more than 1500 IMDs. This classification includes all conditions where the impairment of biochemical pathways is intrinsic to the disorder‘s pathomechanism, while the presence of a diagnostic metabolic biomarker is no longer a prerequisite [4]. In this ever-growing classification, vastly expanded conventional IMD categories such as congenital disorders of glycosylation can be found, as well as recently defined IMD groups such as congenital disorders of autophagy [5], disorders of metabolite repair/proofreading [6], and disorders of the synaptic vesicle cycle [7]. Metabolic epilepsies belong to various IMD groups [8,9]. A significant number of these diseases have specific treatments that may significantly change health outcomes if diagnosed in time. This treatment not only replaces or complements conventional treatment with antiseizure drugs (ASD), but also targets the pathophysiology of the disorder and other systemic symptoms besides seizures. In recent years, the number of treatable metabolic epilepsies has increased significantly due to major breakthroughs in treatment strategies, elucidation and targeting of diseases’ molecular mechanisms and the application of new methodologies for clinical trials in small populations, which are applicable to rare diseases [8,10].

Although the International League Against Epilepsies (ILAE) Commission for Classification and Terminology states in its Position Paper that “An etiologic diagnosis should be considered from when the patient first presents, and at each step along the diagnostic pathway” [1], there is a lack of any classification of monogenic epilepsies, and ILAE recommendations for genetic testing in epilepsy were developed in the pre-genomic era [11]. The ILAE on-line diagnostic manual only provides information on 59 epilepsy genes and 8 IMDs or IMD groups and does not reflect the real scope of monogenic epilepsies [12]. Meanwhile, more than 1000 human genes have been associated with monogenic disorders that involve epilepsy or seizures and a considerable portion of them (373 genes out of a list of 880 genes (42%) in our previous study) are IMDs [13]. Metabolic epilepsies comprise a significant part of the etiologies in certain groups of epilepsy patients, such as neonatal epilepsy [14], refractory epilepsy [15], status epilepticus [16] and progressive myoclonus epilepsies [17], among others. Besides, although it is generally perceived that metabolic epilepsies account for a very small proportion of all patients with epilepsy [1,18], IMDs are frequently diagnosed through genomic testing in various cohorts of patients with epilepsy, including unselected cohorts of patients with variable epilepsy phenotypes, early-onset epilepsy, progressive myoclonic epilepsy and even adult or elderly patients with epilepsy (Table 1).

The main goals of our study were to identify the scope of metabolic epilepsies ac-cording to the current classification of IMDs (ICIMD), and to investigate the clinical presentation, diagnostic approaches and treatments of these metabolic epilepsies.

## 2. Materials and Methods

The International Classification of Inherited Metabolic Disorders (ICIMD) and IEMbase were used as a basis for the identification and classification of metabolic epilepsies [4,23]. All IMDs including epilepsy or seizures as a clinical symptom were included in the list. Information on the clinical presentation and diagnostics of every IMD listed in the ICIMD was obtained mainly from the database, Online Mendelian Inheritance in Man (OMIM; https://www.ncbi.nlm.nih.gov/omim (accessed on 10 December 2021) [23], and missing information was supplemented by literature searches. Clinical synopses of metabolic epilepsies are included in the Appendix A. Information on the treatment of diseases was mostly based on our previous studies [8,24,25] and supplemented by literature searches as required. For the literature search, we used a targeted approach to identify missing information on specific IMD’s clinical presentation, diagnostics and/or treatment. We searched PubMed (http://www.ncbi.nlm.nih.gov/pubmed (accessed on 10 December 2021), restricted searches to English language and publications in peer-reviewed journals, encompassing a period up to December 2021.

The definitions and terms relevant to this study are presented in Table 2.

## 3. Results

### 3.1. The Scope of Metabolic Epilepsies

In this study, 600 metabolic epilepsies have been identified, accounting for as much as 37% of all currently described IMDs (1625 IMDs in the ICIMD and IEMbase, as of 10 December 2021), i.e., epilepsy or seizures are common manifestations of IMDs (Table 3, Figure 1). Epilepsy or seizures are particularly common symptoms in some groups of IMDs: Energy metabolism defects, especially mtDNA-related disorders (30/37 disorders, 81%) and disorders of energy substrate metabolism (19/29, 66%);Some disorders of amino acid metabolism, such as urea cycle disorders and inherited hyperammonemia, organic acidurias and disorders of branched-chain amino acid metabolism (24/33 disorders, 73%) and disorders of glycine and serine metabolism, glutamate/glutamine and aspartate/asparagine metabolism (13/15 disorders, 87%);Complex molecule and organelle metabolism defects, including all three groups—congenital disorders of glycosylation (81/144 disorders, 56%), disorders of organelle biogenesis, dynamics and interactions (60/119 disorders, 51%) and disorders of complex molecule degradation (43/75 disorders, 57%);Neurotransmitter disorders (40/69 disorders, 58%).

Like other IMDs, the majority of metabolic epilepsies are inherited as autosomal recessive (497 diseases, 83% of all metabolic epilepsies), 68 diseases are autosomal dominant (11% of all metabolic epilepsies), 34 metabolic epilepsies are X-linked (6% of all metabolic epilepsies), and 1 metabolic epilepsy (catalytic phosphatidylinositol 3-kinase subunit α superactivity) is caused by a somatic mutation of *PIK3CA* (Appendix A).

### 3.2. Clinical Presentation of Metabolic Epilepsies

Seizures in metabolic epilepsies may present variably. In some diseases, seizures occur in a subset of patients only (e.g., epilepsy is present in about half of individuals affected with SSADH deficiency [26]); in other cases, epilepsy is a constant symptom (e.g., GABA-transaminase deficiency [27]). The age of presentation can be a diagnostic indication in some metabolic epilepsies [8,28]; however, it must be emphasized that there can be a considerable overlap among age groups, many disorders (e.g., mitochondriopathies) may present at any age and there is a general trend for the expansion of the clinical spectrum of many IMDs towards adolescent or adult-onset phenotypes with improved diagnostics [29,30].

Although seizure semiology can be highly variable, the presence of certain types of seizures such as progressive myoclonic seizures [17], infantile spasms [31], epilepsia partialis continua [8] or refractory myoclonic or tonic seizures in the context of a burst suppression pattern on electroencephalogram (EEG) in the neonatal period [14] should raise the suspicion of metabolic epilepsy. Besides, although metabolic epilepsies are more frequently associated with generalized seizures, focal epilepsy may be a symptom in a number of IMDs, especially those associated with cortical malformations (e.g., peroxisomal disorders or congenital disorders of glycosylation) [32]. Encephalopathy is characteristic of 112 IMDs (19%) that are often classified as developmental and epileptic encephalopathies (DEE) (e.g., early infantile epileptic encephalopathies due to *GABRA1, GABRB1, GABRB2* or *GABRB3* pathogenic variants; Appendix A). The effects of treatment with antiseizure drugs (ASD) vary widely; some diseases readily respond to ASD, but at least 91 IMDs (15%) may cause refractory or intractable seizures and status epilepticus. Many disorders of intermediary metabolism present with symptomatic seizures during metabolic crises—at least 40 disorders (7%) may lead to metabolic coma—and persistent seizures may develop as a consequence of these crises.

Most of these metabolic epilepsies are complex disorders and encompass not only seizures but also other symptoms, i.e., the presentation is often multisystemic (Figure 2). Epilepsy is usually accompanied by developmental delay and/or intellectual disability (400 metabolic epilepsies, 66% of all metabolic epilepsies) that may develop after a period of normal development (e.g., in neurodegenerative diseases as disorders of glycosaminoglycan degradation). Other frequent symptoms are muscular hypotonia (387 diseases, 64.5% of all metabolic epilepsies), microcephaly (40.5% of all metabolic epilepsies) and ataxia (155 disorders, 26% of all metabolic epilepsies). Behavioral or psychiatric symptoms occur in at least 126 metabolic epilepsies (21% of all metabolic epilepsies), and dystonia occurs in 100 (17%) of all metabolic epilepsies. Other systems and organs may also be affected, e.g., cardiomyopathy is a symptom of 35 metabolic epilepsies (6% of all metabolic epilepsies), kidney disorders or anomalies are a symptom of 74 metabolic epilepsies (12% of all metabolic epilepsies), hearing impairment or deafness are a symptom of 84 (14% of all metabolic epilepsies). Clinical synopses of metabolic epilepsies are presented in Appendix A.

### 3.3. Diagnostics of Metabolic Epilepsies

In 422 diseases (70% of all metabolic epilepsies), abnormalities can be identified by using routine laboratory tests and/or metabolic testing (Figure 3). Metabolic testing includes a wide range of investigations from conventional tests available in almost every metabolic laboratory (e.g., plasma amino acids, acylcarnitines or urinary organic acids) to highly-specialized (e.g., analysis of plasma sphingolipids for the diagnosis of sphingolipidoses) or invasive investigations (e.g., biopsies of various tissues). In some cases, these abnormalities are not disease-specific (e.g., an increase in plasma alanine concentration in patients with hyperlactacidemia). Abnormalities identified by routine laboratory tests include changes in blood ammonia, acid-base, ketones, lactate or glucose concentrations, changes in complete blood count, among others. One hundred and fourteen diseases (19% of all metabolic epilepsies) are characterized by abnormalities found in metabolic testing only, and 227 diseases (38% of all metabolic epilepsies) are characterized by abnormalities found in both routine and metabolic testing; hence, more or less specific metabolic biomarkers may be identified in 341 metabolic epilepsies (57% of all metabolic epilepsies). In 81 diseases (13.5% of all metabolic epilepsies), only abnormalities in routine laboratory tests are detected.

Of the 110 treatable metabolic epilepsies (Section 3.4. Treatable metabolic epilepsies, below), 61 diseases (55% of treatable metabolic epilepsies) are characterized by changes in both routine and metabolic studies, and 25 diseases (23% of treatable metabolic epilepsies) are characterized by abnormalities in metabolic testing only; thus, more or less specific metabolic biomarkers are characteristic of 86 treatable metabolic epilepsies (78% of treatable metabolic epilepsies) (Figure 3). Abnormalities in routine laboratory tests only are identified in 13 treatable metabolic epilepsies (12% of treatable metabolic epilepsies), and 11 treatable metabolic epilepsies (10% of treatable metabolic epilepsies) do not present any changes in routine laboratory or metabolic tests and can only be diagnosed through molecular genetic testing.

### 3.4. Treatable Metabolic Epilepsies

In this study, 111 metabolic epilepsies (18% of all) are treatable. These diseases have specific treatments that target the pathophysiology of the disease and usually affect not only seizures but also other neurologic and/or systemic symptoms (Table 4). The majority of treatable metabolic epilepsies are amenable to nutritional therapy (65 IMDs, 59% of all treatable metabolic epilepsies), followed by pharmacological therapy (37 IMDs, 34%), vitamin and trace element substitution (36 IMDs, 32%), hemodialysis/peritoneal dialysis (13 IMDs, 12%), solid organ transplantation (12 IMDs, 11%), hematopoietic stem cell transplantation (5 IMDs, 4.5%), gene-based therapy (3 IMDs, 3%), and enzyme replacement therapy (2 IMDs, 2%). In some treatable metabolic epilepsies, more than one treatment strategy is applied. Relatively simple, inexpensive and (often) quite effective nutritional and vitamin/trace element substitution therapies [33] are effective in 59% and 32% of treatable metabolic epilepsies in this study.

The majority of treatable metabolic epilepsies are related to the following three groups of IMDs: defects of intermediary metabolism of nutrients and energy, and cofactor/mineral metabolism defects. Nutritional and vitamin/trace element substitution therapies are mostly applied in these diseases. The list of treatable metabolic epilepsies also includes six congenital disorders of glycosylation (which are amenable to various treatment strategies including nutritional, pharmacological and vitamin/trace element substitution) and seven lysosomal storage diseases that are mostly amenable to the more sophisticated stem cell or gene-based therapies (defects of complex molecule and organelle metabolism). Finally, defects of metabolic cell signaling encompass ten treatable metabolic epilepsies that are mostly amenable to pharmacological treatments, e.g., memantine and/or intravenous immunoglobulins for ionotropic glutamate receptor NMDA-type defects [24].

## 4. Discussion

Although it is commonly thought that metabolic epilepsies account for only a small fraction of all monogenic epilepsies and of all patients with epilepsy, the real scope and frequency of metabolic epilepsies remain underestimated. As many as 600 metabolic epilepsies have been identified in this study, which represent 37% of all currently known IMDs. The number of metabolic epilepsies is likely to increase still further; as human metabolic pathways involve more than 3200 genes, many potential future IMDs remain to be discovered [34]. Recently, several novel categories of IMDs such as congenital disorders of autophagy, disorders of the synaptic vesicle and disorders of metabolite repair/proofreading have been defined. These groups of IMDs may also expand in the future, e.g., the ICIMD currently includes 12 congenital disorders of autophagy [35], while a database of autophagy-associated human genes currently includes 679 genes, http://www.tanpaku.org/autophagy (accessed on 10 December 2020); 163 of these genes are associated with monogenic diseases, while 65 diseases (40%, 65/163) may present with epilepsy or seizures, including many of the so-called mTORopathies where defects of autophagy and mitophagy comprise some of the main pathogenetic mechanisms [36,37]. Autophagy is a fundamental and conserved catabolic pathway that is particularly important to post-mitotic and metabolically active cells such as neurons; hence, the prominent involvement of the central nervous system, including seizures, is a characteristic of congenital disorders of autophagy [38].

The identification and characterization of the genetic landscape of metabolic epilepsies is highly important for several reasons: this knowledge can help achieve the “triple aim” of healthcare organizations, i.e., improvement of patient experiences with care, improvement of health outcomes, and better management of health system impacts through better organization of healthcare services (including diagnostics, treatment and multidisciplinary care) [39]. In addition, research into the molecular mechanisms of metabolic epilepsies may pave the way towards new areas of research and ultimately lead to the discovery of new treatments in both rare and common epilepsies [10,40].

The first steps towards further progress represent well-established principles of effective diagnostics. The establishment of the precise etiology is highly beneficial to patients, families and societies: this not only allows for specific, prognosis-altering treatments, but also has implications for prognosis, a family’s reproductive and life plans, provides guidance for further patient management, allows for engagement with peer-support groups, and halts the diagnostic odyssey, including invasive and expensive testing. Besides, it may provide the means for further investigation of the disease’s molecular mechanisms and the inclusion of patients into clinical trials [41]. Metabolic epilepsies are at the intersection of the disciplines of epileptology, IMDs and genetics, and the referring physician will most often be a metabolic pediatrician/internist, neurologist/epileptologist or geneticist. Conventional investigations used for the diagnosis of epilepsy by epileptologists or neurologists (such as seizure semiology, electrophysiological, or imaging investigations) only rarely provide diagnostic clues for the etiological diagnosis of IMD [13]. Clinical “red flags” for IMD include parental consanguinity or family history for a similar condition, multisystem involvement, abnormalities in routine laboratory testing, fluctuating course of illness, seizures related to fasting, food intake or catabolic stress, unexplained neonatal seizures, refractory seizures or myoclonic seizures, encephalopathic episodes, and neuro-regression [8,42,43]. At least two thirds of metabolic epilepsies result in developmental delay and/or intellectual disability, which is much more common than in the general population of patients with epilepsy (25% of all children with epilepsy experience neurocognitive deficits [44]). However, the whole range of symptoms and signs in metabolic epilepsies is highly diverse, and the symptoms are frequently non-specific and overlapping with non-IMDs (Appendix A). Therefore, IMDs may be diagnosed “unexpectedly” in patients with previously unsuspected IMD [45,46], whereas other genetic or non-genetic diseases can be identified in patients suspected of suffering from IMDs [47]. Moreover, 18% of all metabolic epilepsies are treatable, but early diagnosis is a mandatory prerequisite for successful treatment, and the whole clinical spectrum of the disease may still not be fully developed at the time of diagnosis.

Metabolic epilepsies are diagnosed through metabolic and/or genetic testing methods. Unfortunately, there is no established “gold standard” for one or another testing modality in every clinical situation, and both methods have their own pros and cons. In some cases, metabolic testing has higher specificity and sensitivity and shorter turnaround time, but only non-specific metabolic biomarkers are identified in many IMDs, with the further need for diagnostic confirmation through genetic testing. The non-specificity of metabolic biomarkers may even be misleading (e.g., patients with pyridoxine-dependent epilepsy due to *PLPBP* gene mutations were misdiagnosed with mitochondrial or glycine encephalopathies due to non-specific abnormalities in metabolic testing) [48]. Another drawback of metabolic testing is the need for a diverse spectrum of targeted biochemical assays that analyze a limited number of metabolites each; some of these tests are very rare, highly-specialized and have limited availability [49]. Indeed, when only conventional, more widely available metabolic testing is used for IMD diagnostics, diagnostic yields are usually very low; e.g., the diagnostic yield in a neonatal intensive care unit (190 neonates tested) was only 5.6% [50], even though many IMDs that are not diagnosed through conventional metabolic testing may present in the neonatal period [14]. Similarly, the diagnostic yield of metabolic testing in epileptic encephalopathies (110 patients tested) was only 7% [51]. Some recently developed metabolomic methods provide opportunities for wider and untargeted diagnostics; however, their application in clinical practice is still limited [49,52]. In recent years, genomic testing is replacing single-gene sequencing in clinical practice and provides opportunities for more or less untargeted diagnostics, especially in patients with non-specific symptoms where the probability of diagnosing IMD is lower. However, the specificity and sensitivity may be lower and the turnaround time may be longer for genetic testing, which may result in missed diagnoses [53]. Moreover, there is great variability in genomic testing and in many cases gene panels are used instead of exome or genome sequencing to avoid incidental findings, variants of unknown significance and the burden of the interpretative workload for identified variants. Unfortunately, gene panels differ greatly in both the number and composition of genes, even when used for similar groups of patients with epilepsy, with a high chance of missing diagnoses, especially in non-specific phenotypes with a vast genetic architecture, such as epilepsy [13,54]. Indeed, bearing in mind the wide genetic heterogeneity of metabolic epilepsies that currently encompass at least 600 nosologies, and the heterogeneity and non-specificity of the clinical presentation, omics methods may provide the optimal way to identify the diagnosis in every patient. “Multi-omics” may be the best strategy for the diagnostics of metabolic epilepsies, especially in unsolved cases; however, the price and availability of such testing are still prohibitive.

In day-to-day clinical practice, the choice of diagnostic strategy may depend on several other factors. (1) The availability of highly-specialized clinical and laboratory expertise and diagnostic methods within the healthcare system. Metabolic tests are usually performed in specialized metabolic-testing laboratories and their availability, especially for less frequently used highly-specialized metabolic tests, varies across countries and regions. Metabolic physicians usually have direct contact with metabolic laboratories, while other professionals may not. Besides, there is a lack of adult metabolic specialists in many countries, whereas in some countries the specialty of metabolician does not exist at all [55]. Genetic testing is performed in both clinical and commercial laboratories and is usually more widely available. (2) The clinical presentation and age of a given patient and index of suspicion for IMDs; e.g., metabolicians are more likely to diagnose inherited disorders of intermediary metabolism, epileptologists are more likely to diagnose neurotransmitter disorders, and geneticists are more likely to diagnose syndromic metabolic epilepsies accompanied by intellectual disability and/or malformations. Importantly, the attitudes of different professionals towards the diagnostics of IMD may vary. For example, a recent study found that epileptologists caring for mostly adult patients were significantly less likely to order genetic testing than were physicians caring for mostly pediatric patients (33% vs. 80%) [56]. (3) The acuity and treatability of illness, where timely diagnosis is essential. Many IMDs, especially those of intermediary metabolism, may present with metabolic crises, and many metabolic epilepsies evoke refractory seizures or status epilepticus [15,16]. In many cases, the turn-around time of metabolic testing, especially for conventional metabolic tests, is better; however, accelerated protocols for genomic testing in the intensive care setting (e.g., genome sequencing) are becoming more widely available [57]. (4) The cost-effectiveness and reimbursement of diagnostic tests. In some healthcare systems, even those of developed countries, genomic testing may be underutilized due to limited reimbursement or perceived costliness despite the established fact that comprehensive and more or less untargeted testing modalities, such as exome sequencing, eventually provide overall cost-effectiveness in larger patient populations [58,59,60].

The development of specific treatments for metabolic epilepsies has prospered in recent years. Potentially prognosis-altering specific treatments that are personalized and targeted to the disease pathophysiological mechanisms are becoming available for an increasing number of metabolic epilepsies. In many cases the treatment consists of relatively simple, inexpensive and (often) quite effective nutritional and vitamin/trace element substitution therapies [33] that comprised 59% and 32% of available specific treatments for metabolic epilepsies in this study. Unfortunately, this success has not been accompanied by similar success in the development of therapies for common epilepsies. Despite the advent of many ASDs over the past 50 years and decades of research into the neurobiology of epilepsy, ASDs remain ineffective in 30 to 40% of epilepsy patients and this proportion has remained essentially unchanged over the years [61]. The development of currently used ASDs is mostly based on ictogenesis models, targets either ion channels or neurotransmitter receptors but not epileptogenesis processes, and does not have disease-modifying properties [62,63]. Indeed, the concept of epilepsy as exclusively a channelopathy that has prevailed for many decades may preclude further progress in the study of epileptogenesis [61,64]. Deeper insights into the molecular mechanisms of metabolic epilepsies may pave the way towards novel or more personalized approaches of epileptogenesis-targeted treatments in both rare and common multifactorial epilepsies. The main groups of metabolic epilepsies identified in this study, including defects of energy metabolism, defects of complex molecule and organelle metabolism, and neurotransmitter disorders, may provide insights into future directions for investigation. As metabolic disturbances play a highly important role in epileptogenesis, various metabolic-targeting models have been developed recently for these purposes, including those for the investigation of various bioenergetic processes [61,65,66], mitochondrial functions [67], autophagy [68] and lipid metabolism [69], among others. As autophagy and mitophagy impairments are implicated in the epileptogenesis mechanisms downstream of mTOR hyperactivation in mTORopathies and malformations of cortical development, these processes provide promising novel therapeutic targets [36,37].

### Study Limitations

Although already quite extensive, this study presents an overview of metabolic epilepsies as of 2022 and is based on the recently-established ICIMD classification, which has been widely endorsed by the IMD community. Therefore, IMDs that are currently unacknowledged in the ICIMD nosology or entirely novel groups of IMDs may not be represented in this list of metabolic epilepsies. Indeed, several novel IMD groups have emerged in recent years (e.g., congenital disorders of autophagy and disorders of the synaptic vesicle) and we foresee the further appearance of such novel groups as we gain deeper insights into the pathomechanisms of many rare and common epilepsies. Constant revision and improvement are intrinsic to the ICIMD and we encourage further studies and revisions in the field of metabolic epilepsies in the coming years.

## 5. Conclusions

Characterization of the genetic architecture of metabolic epilepsies is of paramount importance, both to current clinical practice and for the identification of further research directions.

## Figures and Tables

**Figure 1 genes-13-00508-f001:**
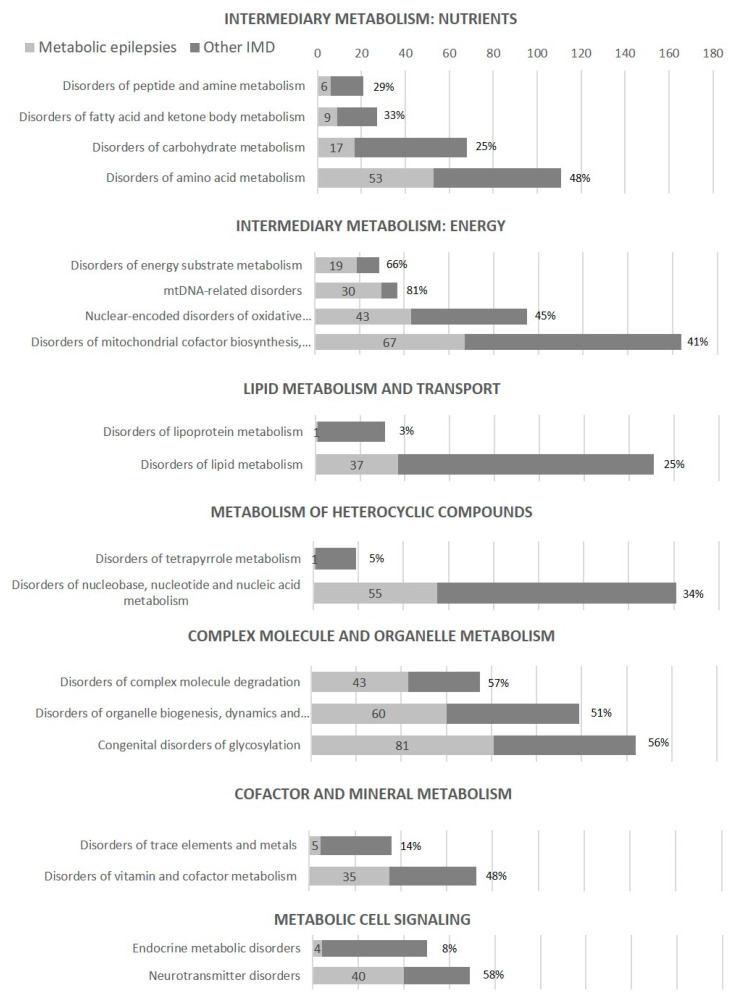
Number and percentage of metabolic epilepsies in ICIMD groups.

**Figure 2 genes-13-00508-f002:**
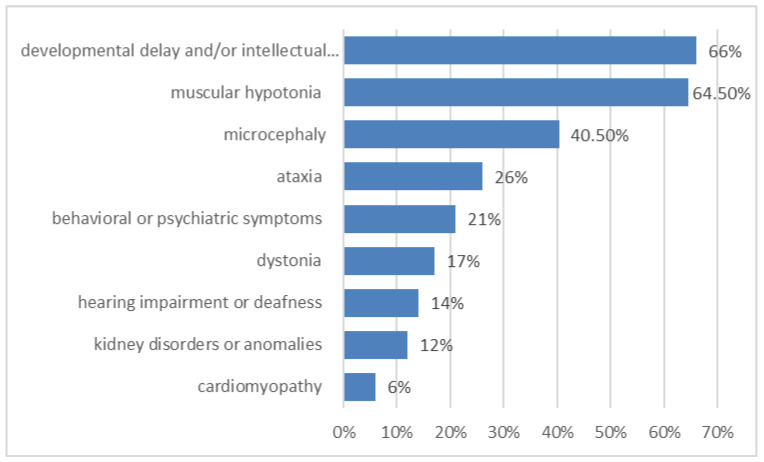
Some of the most frequent accompanying symptoms in metabolic epilepsies (in %).

**Figure 3 genes-13-00508-f003:**
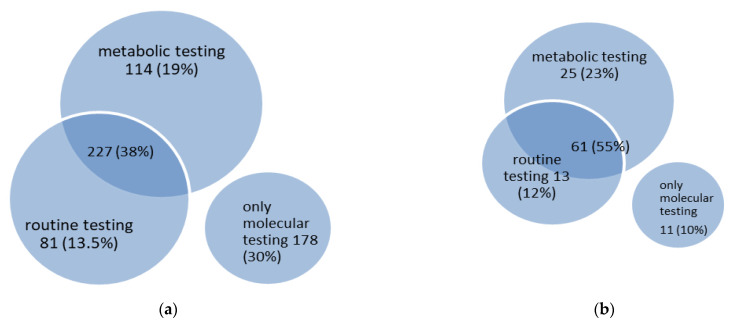
Venn diagram of diagnostic testing in 600 metabolic epilepsies (**a**) and 111 treatable metabolic epilepsies (**b**).

**Table 1 genes-13-00508-t001:** Percentage of diagnosed IMDs in various cohorts of epilepsy patients.

Reference	Cohort of Patients	Percentage of IMDs among All Diagnoses	Diagnostic Method
[19]	197 children with epilepsy, abnormal metabolic investigations was an exclusion criterion	13%	ES
[20]	293 patients with variable epilepsy phenotypes	21.5%	ES
[13]	patients with variable epilepsy phenotypes	38%	TruSight One panel, Illumina
[21]	180 patients with early-life epilepsy (onset ≤ 5 years)	25%	Gene panel of 620 known epilepsy genes
[17]	38 patients with progressive myoclonic epilepsies	62.5%	ES
[22]	150 adult and elderly patients with intellectual disability and epilepsy	18%	TruSight One panel, Illumina/ES

ES—exome sequencing.

**Table 2 genes-13-00508-t002:** Terms and definitions.

Inherited metabolic disorder (IMD) was defined as any primary genetic condition in which alteration of a biochemical pathway is intrinsic to specific biochemical, clinical and/or pathophysiological features. The presence of an abnormal metabolite is no longer a prerequisite [4].
A treatable IMD was defined as the availability of a particular therapeutic modality capable of preventing or improving disease phenotype, i.e., positively influencing the outcome measures [8].
The concept of a metabolic epilepsy is that it results directly from an IMD in which seizures are a core symptom of the disorder [1].
Routine laboratory tests were defined as tests in blood and/or urine that are readily available in regular hospital laboratories in most developed countries.
Metabolic tests were defined as tests that are usually performed in specialized laboratories for IMD diagnostics or require other specialized laboratory investigations (e.g., pathological investigations in biopsies).

**Table 3 genes-13-00508-t003:** Metabolic epilepsies according to the International Classification of Inherited Metabolic Disorders (ICIMD).

Groups of IMDs and Genes	Number and Percentage of Metabolic Epilepsies vs. Number of All Described IMDs in the ICIMD Group
INTERMEDIARY METABOLISM: NUTRIENTS	
**Disorders of amino acid metabolism**	53/111 (48%)
*Urea cycle disorders and inherited hyperammonemias, organic acidurias and disorders of branched-chain amino acid metabolism*	24/33 (73%)
*NAGS, CPS1, OTC, ASS1, ASL, ARG1, SLC25A15, GLUD1, IVD, ACAD8, ACADSB, MCCC1, MCCC2, AUH, ECHS1, HIBCH, PCCA, PCCB, GCDH, MLYCD, BCKDHA, BCKDHB, DBT, BCKDK*	
*Disorders of glycine and serine metabolism, glutamate/glutamine and aspartate/asparagine metabolism*	13/15 (87%)
*GLDC, AMT, PHGDH, PSAT1, PSPH, SLC1A4, GPT2, GAD1, GLUL, GLS, ASNS, NAT8L, ASPA*	
*Other disorders of amino acid metabolism (phenylalanine and tyrosine, sulfur-containing amino acids and hydrogen sulfide, ornithine, proline and hydroxyproline, amino acid transport)*	15/61 (25%)
*PAH, ADK, MTR, CBS, SQOR, ETHE1, SUOX, PYCR2, PRODH, ALDH4A1, AASS, SLC6A19, SLC1A3, SLC1A2, SLC6A1, ACY1*	
**Disorders of peptide and amine metabolism**	6/21 (29%)
*GSS, XPNPEP3, ODC1, SMS*	
**Disorders of carbohydrate metabolism**	17/68 (25%)
*ALDOB, GLYCTK, FBP1, PC, PCK1, GK, HK1, GCK, PGK1, GYS1, GYS2, EPM2A, NHLRC1, RPIA, SLC2A1, SLC45A1, SLC17A5*	
**Disorders of fatty acid and ketone body metabolism**	9/27 (33%)
*CPT1A, CPT2, SLC25A20, ACADS, ACADM, HADH, HADHA, HMGCL, SLC16A1*	
**INTERMEDIARY METABOLISM: ENERGY**	
**Disorders of energy substrate metabolism**	19/29 (66%)
*PDHA1, PDHB, DLAT, DLD, PDHX, PDP1, MPC1, ACO2, IDH2, IDH3A, SUCLA2, SUCLG1, FH, MDH2, SLC13A5, OGDHL, GATM, GAMT, SLC6A8*	
**mtDNA-related disorders**	30/37 (81%)
*MT-ND1, MT-ND2, MT-ND3, MT-ND4, MT-ND5, MT-ND6, MT-CYB, MT-CO1, MT-CO2, MT-CO3, MT-ATP6, MT-ATP8, MT-TR, MT-TN, MT-TC, MT-TQ, MT-TH, MT-TI, MT-TL1, MT-TL2, MT-TK, MT-TM, MT-TF, MT-TP, MT-TS1, MT-TS2, MT-TT, MT-TW, MT-TY, MT-TV*	
**Nuclear-encoded disorders of oxidative phosphorylation**	43/95 (45%)
*NDUFV1, NDUFV2, NDUFS1, NDUFS4, NDUFS6, NDUFS7, NDUFS8, NDUFA1, NDUFA2, NDUFA8, NDUFA11, NDUFB8, NDUFB11, NDUFC2, NDUFAF2, NDUFAF3, NDUFAF4, NDUFAF5, NDUFAF6, NDUFAF8, FOXRED1, NUBPL, TIMMDC1, SDHA, SDHD, UQCC2, CYC1, HCCS, BCS1L, COX4I1, COX6B1, COX8A, NDUFA4, COX10, COX15, SCO2, LRPPRC, PET100, FASTKD2, APOPT1, ATP5F1A, ATP5F1D, TMEM70*	
**Disorders of mitochondrial cofactor biosynthesis, mitochondrial DNA maintenance and replication, mitochondrial gene expression, other disorders of mitochondrial function**	67/164 (41%)
*PDSS2, COQ2, COQ4, COQ5, COQ6, COQ8A, COQ9, LIPT2, LIAS, BOLA3, IBA57, ISCA1, NFS1, DGUOK, RRM2B, SAMHD1, POLG, POLG2, TWNK, HSD17B10, TRNT1, MTFMT, GTPBP3, MTO1, TRIT1, RARS2, NARS2, CARS2, EARS2, IARS2, LARS2, FARS2, PARS2, VARS2, WARS2, KARS2, MRPL12, MRPS22, MRM2, RMND1, GFM1, GFM2, TSFM, GUF1, SLC25A1, SLC25A10, SLC25A12, GOT2, MDH1, SLC25A22, MICU1, TIMM50, PMPCB, MIPEP, CLPB, CLPP, LONP1, HSPD1, FBXL4, AFG3L2, ATAD3A, HTRA2, PPA2, TXN2, AIFM1, RTN4IP1, PTRH2*	
**INTERMEDIARY METABOLISM: OTHERS**	
**Disorders of metabolite repair/proofreading**	3/4 (75%)
*D2HGDH, L2HGDH, ACSF3*	
**LIPID METABOLISM AND TRANSPORT**	
**Disorders of lipid metabolism**	37/151 (25%)
*ELOVL4, ABCD1, ACOX1, HSD17B4, BSCL2, CHKB, PCYT2, MBOAT7, PLA2G6, DDHD2, MFSD2A, FIG4, OCRL, SYNJ1, PIK3CA, PIK3R2, PI4K2A, PTEN, INPP5K, PLCB1, PLCH1, PEX7, FAR1, PEX5, CERS1, DEGS1, FA2H, SGPL1, FDFT1, LSS, NSDHL, EBP, DHCR24, DHCR7, CYP27A1, AMACR*	
**Disorders of lipoprotein metabolism**	1/31 (3%)
*VLDLR*	
**METABOLISM OF HETEROCYCLIC COMPOUNDS**	
**Disorders of nucleobase, nucleotide and nucleic acid metabolism**	55/161 (34%)
*Disorders of pyrimidine, purine, ectonucleotide and nucleic acid metabolism, disorders of ribosomal biogenesis*	23/117 (20%)
*CAD, DPYD, DPYS, UPB1, PRPS1, ADSL, ATIC, AMPD2, ADA2, ITPA, TREX1, RNASEH2A, RNASEH2B, RNASEH2C, RNASET2, ADARB1, IFIH1, POLR3A, POLR3B, UBTF, SNORD118, EMG1, RPL10*	
*Disorders of non-mitochondrial tRNA processing and aminoacyl-tRNA synthetases*	32/44 (73%)
*TSEN2, TSEN15, TSEN34, TSEN54, CLP1, TRMT10A, TRMT1, DALRD3, NSUN2, ADAT3, LAGE3, OSGEP, TP53RK, TPRKB, PUS3, WDR4, AARS1, RARS1, NARS1, CARS1, QARS1, EPRS1, IARS1, LARS1, KARS1, FARSB, SARS1, YARS1, VARS1, AIMP1, AIMP2, NUP133*	
**Disorders of tetrapyrrole metabolism**	1/19 (5%)
*HMBS*	
**COMPLEX MOLECULE AND ORGANELLE METABOLISM**	
**Congenital disorders of glycosylation**	81/144 (56%)
*Disorders of N-linked and O-linked protein glycosylation*	20/33 (61%)
*PMM2, DPAGT1, ALG13, ALG14, ALG1, ALG2, ALG11, RFT1, ALG3, ALG9, ALG6, STT3A, STT3B, OSTC, SSR4, MOGS, MAN1B1, MGAT2, FUT8, FCSK, POMT1, POMT2, POMGNT1, B3GALNT2, POMK, FKTN, FKRP, B4GAT1, EXT2, EXTL3, NDST1, HS6ST2*	
*Disorders of lipid glycosylation*	25/27 (93%)
*PIGA, PIGC, PIGQ, PIGH, PIGP, PIGY, PIGL, PIGW, PIGM, PIGV, PIGN, PIGB, PIGO, PIGF, PIGG, PIGT, PIGS, PIGU, PIGK, GPAA1, PGAP1, PGAP3, PGAP2, ST3GAL5, ST3GAL3*	
*Disorders of multiple glycosylation pathways, other disorders of glycan metabolism*	24/36 (67%)
*DHDDS, NUS1, DOLK, DPM1, DPM2, DPM3, MPDU1, SLC35A1, SLC35A2, SLC35A3, SLC35C1, ATP6V0A2, ATP6V1A, ATP6AP1, ATP6AP2, CCDC115, TMEM165, GNE, NANS, PGM1, GMPPB, UGDH, UGP2, NGLY1*	
**Disorders of organelle biogenesis, dynamics and interactions**	60/119 (51%)
*SERAC1, PISD, MICOS13, DNM1L, MFF, SPATA5, STAT2, SLC25A46, TRAK1, PEX1, PEX2, PEX3, PEX5, PEX6, PEX10, PEX12, PEX13, PEX19, PEX26, VPS11, AP3D1, LYST, MYO5A, RAB27A, BCAP31, COL4A3BP, VPS13A, VPS13B, COG2, COG4, COG4, COG5, COG6, COG7, COG8, ARCN1, TRAPPC2L, TRAPPC4, TRAPPC6B, TRAPPC9, TRAPPC11, TRAPPC12, GOSR2, VPS4A, VPS41, YIF1B, TANGO2, SCYL2, STX11, STXBP2, ARFGEF2, AP1S2, AP3B2, AP4B1, AP4E1, AP4M1, AP4S1, RUBCN, RAB18, AP1G1*	
**Disorders of complex molecule degradation**	43/75 (57%)
*Disorders of sphingolipid degradation*	11/17 (65%)
*GBA, SMPD4, HEXA, HEXB, GM2A, GALC, ARSA, SUMF1, GLA, ASAH1, PSAP*	
*Disorders of glycosaminoglycan degradation, Disorders of glycoprotein degradation, and Other disorders of complex molecule degradation*	14/33 (42%)
*IDS, SGSH, NAGLU, HGSNAT, GNS, NEU1, CTSA, MANBA, NAGA, FUCA1, AGA, SCARB2, NPC1, NPC2*	
*Neuronal ceroid lipofuscinosis*	12/13 (93%)
*PPT1, TPP1, CLN3, DNAJC5, CLN5, CLN6, MFSD8, CLN8, CTSD, ATP13A2, CTSF, KCTD7*	
*Disorders of autophagy*	6/12 (50%)
*EPG5, WDR45, SNX14, SPG11, TECPR2, TBCK*	
**COFACTOR AND MINERAL METABOLISM**	
**Disorders of vitamin and cofactor metabolism**	35/73 (48%)
*GCH1, PTS, SPR, QDPR, SLC19A2, SLC19A3, TPK1, NADK2, NAXD, NAXE, NNT, PANK2, SLC25A42, ALDH7A1, PNPO, PLPBP, ALPL, BTD, HLCS, SLC5A6, SLC46A1, FOLR1, MTHFR, MTHFD1, MTHFS, DHFR, LMBRD1, MMACHC, MMADHC, MMAA, MTRR, HCFC1, MOCS1, MOCS2, GPHN*	
**Disorders of trace elements and metals**	5/36 (14%)
*ATP7A, SLC33A1, FTL, SLC39A8, SEPSECS*	
**METABOLIC CELL SIGNALING**	
**Neurotransmitter disorders**	40/69 (58%)
*Disorders of monoamine neurotransmission, γ-aminobutyric acid neurotransmitter disorders, glutamate neurotransmitter disorders*	19/32 (60%)
*TH, DBH, ABAT, ALDH5A1, GABRA1, GABRB1, GABRB2, GABRB3, GABRG2, GABBR2, GRIN1, GRIN2A, GRIN2B, GRIN2D, GRIA2, GRIA3, GRIA4, ATAD1, GRM1*	
*Disorders of the synaptic vesicle cycle*	18/29 (62%)
*TBC1D24, KIF1A, KIF5A, KIF5C, DYNC1H1, DNM1, PRRT2, SNAP25, SNAP29, STXBP1, SV2A, VAMP2, STX1B, SYN1, IL1RAPL1, DNAJC6, CLTC, DLG4*	
**Endocrine metabolic disorders**	4/50 (8%)
*ABCC8, KCNJ11, AKT2, MC2R*	

Note: in bold—ICIMD groups, in italic—ICIMD subgroups.

**Table 4 genes-13-00508-t004:** Treatable metabolic epilepsies.

Name of the Disorder	Genes	MIM# Number	Treatment Strategy
INTERMEDIARY METABOLISM: NUTRIENTS			
N-acetylglutamate synthase deficiency	*NAGS*	# 237310	Nutritional, pharmacological, solid organ transplantation, hemodialysis/peritoneal dialysis
Carbamoyl phosphate synthetase 1 deficiency	*CPS1*	# 237300	Nutritional, pharmacological, solid organ transplantation, hemodialysis/peritoneal dialysis
Ornithine transcarbamylase deficiency	*OTC*	# 311250	Nutritional, pharmacological, solid organ transplantation, hemodialysis/peritoneal dialysis
Argininosuccinate synthetase deficiency	*ASS1*	# 215700	Nutritional, pharmacological, solid organ transplantation, hemodialysis/peritoneal dialysis
Argininosuccinate lyase deficiency	*ASL*	# 207900	Nutritional, pharmacological, solid organ transplantation, hemodialysis/peritoneal dialysis
Arginase deficiency	*ARG1*	# 207800	Nutritional, pharmacological, solid organ transplantation, hemodialysis/peritoneal dialysis
Mitochondrial ornithine transporter deficiency	*SLC25A15*	# 238970	Nutritional, pharmacological, solid organ transplantation, hemodialysis/peritoneal dialysis
Isovaleryl-CoA dehydrogenase deficiency	*IVD*	# 243500	Nutritional, pharmacological
Mitochondrial short-chain enoyl-CoA hydratase 1 deficiency	*ECHS1*	# 616277	Nutritional
3-hydroxyisobutyryl-CoA hydrolase deficiency	*HIBCH*	# 250620	Nutritional
Propionic acidemia due to propionyl-CoA carboxylase subunit α or β deficiency	*PCCA, PCCB*	# 606054	Nutritional, pharmacological, solid organ transplantation, hemodialysis/peritoneal dialysis
Glutaryl-CoA dehydrogenase deficiency	*GCDH*	# 231670	Nutritional
Branched-chain ketoacid dehydrogenase E1 α/E1 β/dihydrolipoyl transacylase deficiency	*BCKDHA, BCKDHB, DBT*	# 248600	Nutritional, vitamin or trace element, hemodialysis/peritoneal dialysis, solid organ transplantation
Branched-chain ketoacid dehydrogenase kinase deficiency	*BCKDK*	# 614923	Nutritional
Phenylalanine hydroxylase deficiency	*PAH*	# 261600	Nutritional, pharmacological, enzyme replacement
Methionine synthase deficiency	*MTR*	# 250940	Vitamin or trace element
Cystathionine β-synthase deficiency	*CBS*	# 236200	Nutritional, vitamin or trace element
Nonketotic hyperglycinemia due to glycine decarboxylase/aminomethyltransferase deficiency	*GLDC, AMT*	# 605899	Pharmacological
3-phosphoglycerate dehydrogenase deficiency	*PHGDH*	# 601815	Nutritional
Phosphoserine aminotransferase deficiency	*PSAT1*	# 610992	Nutritional
Phosphoserine phosphatase deficiency	*PSPH*	# 614023	Nutritional
Glutamine synthetase deficiency	*GLUL*	# 610015	Nutritional
3-hydroxy-3-methylglutaryl-CoA lyase deficiency	*HMGCL*	# 246450	Nutritional
**INTERMEDIARY METABOLISM: ENERGY**			
GLUT1 deficiency	*SLC2A1*	# 606777; # 612126	Nutritional, pharmacological
Pyruvate dehydrogenase E1 α/E1 β/E3-binding protein/dihydrolipoamide acetyltransferase/dihydrolipoamide dehydrogenase deficiency	*PDHA1, PDHB, DLAT, DLD, PDHX*	# 312170; # 614111; # 245348; # 246900; # 245349	Nutritional, vitamin or trace element
Pyruvate dehydrogenase phosphatase deficiency	*PDP1*	# 608782	Nutritional
Arginine:glycine amidinotransferase (AGAT) deficiency	*GATM*	# 612718	Nutritional
Guanidinoacetate methyltransferase deficiency	*GAMT*	# 612736	Nutritional
Creatine transporter deficiency	*SLC6A8*	# 300352	Nutritional
NADH dehydrogenase core subunit 1/subunit 4/subunit 5/subunit 6/cytochrome c oxidase subunit 1/ mitochondrial tRNA-Gln/tRNA-His/tRNA-Leu 1/tRNA-Phe/tRNA-Ser 1/tRNA-Ser 2/tRNA-Trp deficiency	*MT-ND1, MT-ND4, MT-ND5, MT-ND6, MT-CO1, MT-TQ, MT-TH, MT-TL1, MT-TF, MT-TS1, MT-TS2, MT-TW*	# 540000	Nutritional
Coenzyme Q5 methyltransferase/Q8A (ADCK3) deficiency	*COQ5, COQ8A*	# 619028; # 612016	Vitamin or trace element
Mitochondrial aspartate-glutamate carrier isoform 1 deficiency	*SLC25A12*	# 612949	Nutritional
Mitochondrial aspartate aminotransferase deficiency	*GOT2*	# 618721	Nutritional, vitamin or trace element
**LIPID METABOLISM AND TRANSPORT**			
X-linked adrenoleukodystrophy	*ABCD1*	# 300100	Gene-based, stem cell
7-dehydrocholesterol reductase deficiency	*DHCR7*	# 270400	Nutritional, pharmacological
Sterol 27-hydroxylase deficiency	*CYP27A1*	# 213700	Pharmacological
**METABOLISM OF HETEROCYCLIC COMPOUNDS**			
CAD trifunctional protein deficiency	*CAD*	# 616457	Pharmacological
Phosphoribosylpyrophosphate synthetase deficiency	*PRPS1*	# 301835	Pharmacological
Isoleucyl-tRNA synthetase 1 deficiency	*IARS1*	# 617093	Nutritional
Leucyl-tRNA synthetase 1 deficiency	*LARS1*	# 615438	Nutritional
Phenylalanyl-tRNA synthetase subunit β deficiency	*FARSB*	# 613658	Nutritional
Seryl-tRNA synthetase 1 deficiency	*SARS1*	# 617709	Nutritional
**COMPLEX MOLECULE AND ORGANELLE METABOLISM**			
PMM2-CDG	*PMM2*	# 212065	Pharmacological
PIGA-CDG	*PIGA*	# 300868	Nutritional
PIGM-CDG	*PIGM*	# 610293	Pharmacological
PIGO-CDG	*PIGO*	# 614749	Vitamin or trace element
SLC35A2-CDG	*SLC35A2*	# 300896	Nutritional
SLC35C1-CDG	*SLC35C1*	# 266265	Nutritional
Arylsulfatase A deficiency	*ARSA*	# 250100	Gene-based, stem cell
Iduronate sulfatase deficiency	*IDS*	# 309900	Stem cell
α-fucosidase deficiency	*FUCA1*	# 230000	Stem cell
Aspartylglucosaminidase deficiency	*AGA*	# 208400	Stem cell
Tripeptidyl-peptidase 1 deficiency	*TPP1*	# 204500	Enzyme replacement
CLN7 disease	*MFSD8*	# 610951	Gene-based
Niemann–Pick disease type C1/type C2	*NPC1, NPC2*	# 257220; # 607625	Pharmacological
**COFACTOR AND MINERAL METABOLISM**			
Autosomal recessive GTP cyclohydrolase 1 deficiency	*GCH1*	# 233910	Nutritional, vitamin or trace element, pharmacological
Sepiapterin reductase deficiency	*SPR*	# 612716	Pharmacological, vitamin or trace element
Dihydropteridine reductase deficiency	*QDPR*	# 261630	Nutritional, pharmacological, vitamin or trace element
Thiamine transporter 2 deficiency	*SLC19A3*	# 607483	Vitamin or trace element
Thiamine pyrophosphokinase deficiency	*TPK1*	# 614458	Vitamin or trace element
NAD(P)HX epimerase deficiency	*NAXE*	# 617186	Vitamin or trace element
α-aminoadipic semialdehyde dehydrogenase deficiency	*ALDH7A1*	# 266100	Vitamin or trace element, nutritional
Pyridoxamine 5′-phosphate oxidase deficiency	*PNPO*	# 610090	Vitamin or trace element
PROSC deficiency	*PLPBP*	# 617290	Vitamin or trace element
Biotinidase deficiency	*BTD*	# 253260	Vitamin or trace element
Holocarboxylase synthetase deficiency	*HLCS*	# 253270	Vitamin or trace element
Sodium-dependent multivitamin transporter deficiency	*SLC5A6*	# 618973	Vitamin or trace element
Proton-coupled folate transporter deficiency	*SLC46A1*	# 229050	Vitamin or trace element
Folate receptor α deficiency	*FOLR1*	# 613068	Vitamin or trace element
5,10-methylenetetrahydrofolate reductase deficiency	*MTHFR*	# 236250	Nutritional, vitamin or trace element
5,10-methenyltetrahydrofolate synthetase deficiency	*MTHFS*	# 618367	Vitamin or trace element
Dihydrofolate reductase deficiency	*DHFR*	# 613839	Vitamin or trace element
Methylmalonic aciduria and homocystinuria, cblF type	*LMBRD1*	# 277380	Vitamin or trace element
Methylmalonic aciduria and homocystinuria, cblC type, cblD type	*MMACHC, MMADHC*	# 277400; # 277410	Nutritional, vitamin or trace element
Methylmalonic aciduria, cblA type	*MMAA*	# 251100	Nutritional, vitamin or trace element, pharmacological, hemodialysis/peritoneal dialysis, solid organ transplantation
Methionine synthase reductase deficiency	*MTRR*	# 236270	Vitamin or trace element
Cyclic pyranopterin monophosphate synthase deficiency	*MOCS1*	# 252150	Pharmacological
Copper-transporting ATPase subunit α deficiency	*ATP7A*	# 309400	Pharmacological, vitamin or trace element
SLC39A8 deficiency	*SLC39A8*	# 616721	Nutritional
**METABOLIC CELL SIGNALING**			
Tyrosine hydroxylase deficiency	*TH*	# 605407	Pharmacological
Succinic semialdehyde dehydrogenase deficiency	*ALDH5A1*	# 271980	Pharmacological
Ionotropic glutamate receptor NMDA type subunit 1 dysregulation	*GRIN1*	# 617820; # 614254	Pharmacological
Ionotropic glutamate receptor NMDA type subunit 2A dysregulation	*GRIN2A*	# 245570	Pharmacological
Ionotropic glutamate receptor NMDA type subunit 2B dysregulation	*GRIN2B*	# 613970	Nutritional
Ionotropic glutamate receptor NMDA type subunit 2D superactivity	*GRIN2D*	# 617162	Pharmacological
ATP-sensitive potassium channel regulatory subunit superactivity	*ABCC8*	# 256450	Pharmacological
ATP-sensitive potassium channel pore-forming subunit superactivity	*KCNJ11*	# 618856	Pharmacological
AKT2 superactivity	*AKT2*	# 240900	Pharmacological
ACTH receptor deficiency	*MC2R*	# 202200	Pharmacological

## Data Availability

Not applicable.

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
