# Peer review of "2022 Overview of Metabolic Epilepsies"

_genes, 2022, doi:10.3390/genes13030508_

Round 1

Reviewer 1 Report

This article provides an overview of the landscape of metabolic epilepsies and aim to provide a summary of the diagnostic and treatment considerations for various metabolic epilepsies. This paper is highly significant and timely; and will provide a great tool for other researchers in the field as a brief yet comprehensive summary of metabolic epilepsy. I would like to thank the authors for providing the field with such a paper.

That being said, I have a couple of concerns about the paper:

  1. My biggest concern with this study is the limitation of the approach in the study. By using the database of the ICIMD to classify 'metabolic epilepsy', there is a bias towards only characterising 'metabolic epilepsy' from among the classic IMDs classification. Therefore, there are many epilepsy syndromes that I would classically call 'metabolic epilepsy' that are not included in this study or list. Such examples are the mTORopathies like the TSC or PTEN disorders; which are linked to mTOR signalling, a classic metabolic signalling pathway, and yet not represented at all among the metabolic epilepsy list. Another concern that I would like the author to clarify is that I believe this list overlooks mitochondrial genes. While there are mitochondrial disease genes listed in the list like POLG; the classic mitochondrial disease mutations like the m3243.A>G mutation in the MTTL1 gene that causes MELAS are not represented. Among the 'mitochondrial disease' group, certainly MELAS or MERRF are probably the more classic metabolic epilepsy disorders and yet are unrepresented in the list. These limitations should be acknowledged by the authors and at the very least discussed in the discussions section to acknowledge that there are other 'metabolic epilepsies' out there not represented in this study.
  2. This paper also bases the definition of 'metabolic epilepsies' from a very clinical and classic genetic/inherited perspective. It would be good to hear the author's perspective on how gene discoveries in preclinical studies that could inform our understanding of mechanisms underlying 'metabolic epilepsies' may be underrepresented in this list. For example, among the TSC and other focal epilepsy field, the importance of circadian genes in preclinical studies are gaining traction. While this may not be the primary mechanism for these epilepsies, their modulation of metabolism and its contribution to epilepsy may be as important and may constitute the larger scope of 'metabolic epilepsy'. Perhaps this can be discussed in the discussion section. 
  3. For section 3.2, if the data is available, there should be a discussion of the epilepsy semiology among the list of IMDs. How many of these are focal vs generalised epilepsy? If focal, is there specific brain areas that are represented. What are the typical age that these seizure manifests?
  4. For section 3.4, it would be interesting to define out of the groups in table 2; which disease groups are most treatable or not. The discussion of this would relate to line 238. Perhaps, the authors can use this information to say where more research needs to be done among the 'metabolic epilepsy' field.

Some specific minor concerns of mine below :

  1. Some of the description of the results are very hard to read and understand. For example: line 70-91 are very dense and hard to compare - perhaps a table to summarise the proportion and percentage from these various studies?
  2. Description of many of the results are not ordered from the highest to the lowest; making appreciation of the trend in the data a bit harder. I suggest the following lines to be revised to be in order from highest to lowest: lines 117-129, lines 158-167. 
  3. Figure 1 and Figure 1 should be reordered to make the charts in order from highest to lowest; similar to point 2. Also for Figure 1; perhaps a numerical percentage at the end of each chart would make interpretation of each bar chart easier?
  4. Line 148 and 150 - percentage should also be there next to the number?
  5. Line 51 - should be systemic instead of systematic?
  6. Line 104 - 'supplemented by literature searches as required'?
  7. Line 193 - author should clarify if 'do not present with any changes in routine laboratory or metabolic tests' mean there is no diagnostic markers or that the disease needs molecular testing to diagnose.
  8. Line 220 - 'IMDs such as congenital disorders ....'

Author Response

Reviewer 1:

This article provides an overview of the landscape of metabolic epilepsies and aim to provide a summary of the diagnostic and treatment considerations for various metabolic epilepsies. This paper is highly significant and timely; and will provide a great tool for other researchers in the field as a brief yet comprehensive summary of metabolic epilepsy. I would like to thank the authors for providing the field with such a paper.

That being said, I have a couple of concerns about the paper:

My biggest concern with this study is the limitation of the approach in the study. By using the database of the ICIMD to classify 'metabolic epilepsy', there is a bias towards only characterising 'metabolic epilepsy' from among the classic IMDs classification. Therefore, there are many epilepsy syndromes that I would classically call 'metabolic epilepsy' that are not included in this study or list. Such examples are the mTORopathies like the TSC or PTEN disorders; which are linked to mTOR signalling, a classic metabolic signalling pathway, and yet not represented at all among the metabolic epilepsy list.

Thank you very much for this very relevant remark and we would like to stress that we share and agree with it very much. We also tried to highlight this in the manuscript stating that the real scope and frequency of metabolic epilepsies may currently remain underestimated. Moreover, the number of metabolic epilepsies is likely to increase still further, as mentioned in our example of a very recently described group of congenital disorders of autophagy that currently includes only 12 genes but will presumably grow still further, as at least 679 autophagy–associated genes are currently known (paragraph 214-230). We provide still further stress on that based on your suggestion in Discussion. Having said that, we would like to explain several reasons to take the ICIMD classification as a basis for this study: 1) unfortunately, there is no any other comparably extensive and up-to-date classification for either genetic epilepsies or IMDs that could be taken as a basis for the identification of metabolic epilepsies; 2) luckily, the ICIMD classification already provides a revolutionary approach to IMD definition and classification and represents a huge expansion of the IMD field with many novel disease categories now ascribed to IMDs that were previously not considered metabolic (e.g., disorders of the synaptic vesicle cycle or congenital disorders of autophagy). After the first proposals and some discussions among the major leaders in the field [https://pubmed.ncbi.nlm.nih.gov/29884839/; https://pubmed.ncbi.nlm.nih.gov/30883825/], the final consensus for the establishment of ICIMD was finally reached among the major professional and other IMD organizations; 3) as it is stated by the establishers of the ICIMD classification, “by necessity, the classification is subject to constant revision as a result of the rapid pace of progress in the field of genetics” (http://www.iembase.org/iem-classification.asp); procedures for constant revisions and leverage on the extensive panel of experts provide means for a constant improvement. Therefore, the manuscript may only state that this overview of metabolic epilepsies pertains to the situation in 2022 and acknowledge that further revisions and improvements are very much welcome in the coming years. Finally, we include a paragraph on Limitations of the study to further acknowledge this limitation.

Another concern that I would like the author to clarify is that I believe this list overlooks mitochondrial genes. While there are mitochondrial disease genes listed in the list like POLG; the classic mitochondrial disease mutations like the m3243.A>G mutation in the MTTL1 gene that causes MELAS are not represented. Among the 'mitochondrial disease' group, certainly MELAS or MERRF are probably the more classic metabolic epilepsy disorders and yet are unrepresented in the list. These limitations should be acknowledged by the authors and at the very least discussed in the discussions section to acknowledge that there are other 'metabolic epilepsies' out there not represented in this study.

Thank you, we have corrected this in the Table 2 and Figure 1 where it was accidentally omitted in the first draft of the manuscript. This group of disorders has been included into Table 3 in the first version of the manuscript.

This paper also bases the definition of 'metabolic epilepsies' from a very clinical and classic genetic/inherited perspective. It would be good to hear the author's perspective on how gene discoveries in preclinical studies that could inform our understanding of mechanisms underlying 'metabolic epilepsies' may be underrepresented in this list. For example, among the TSC and other focal epilepsy field, the importance of circadian genes in preclinical studies are gaining traction. While this may not be the primary mechanism for these epilepsies, their modulation of metabolism and its contribution to epilepsy may be as important and may constitute the larger scope of 'metabolic epilepsy'. Perhaps this can be discussed in the discussion section.

Again, thank you very much for this very relevant remark that we tried to address, at least partially, in the final paragraph of Discussion, stating that metabolic disturbances play a highly important role in the epileptogenesis itself and some highly relevant works towards the investigation of these processes in various models are already being performed. Following your suggestion, we include some examples on rare epilepsies and mTORopathies that have a metabolic pathogenetic component. Besides, we include a paragraph on the Limitations of the study to further acknowledge this limitation.

For section 3.2, if the data is available, there should be a discussion of the epilepsy semiology among the list of IMDs. How many of these are focal vs generalised epilepsy? If focal, is there specific brain areas that are represented. What are the typical age that these seizure manifests?

Thank you for the very relevant remark. Unfortunately, the numerical data for both semiology and the age of presentation of all 600 metabolic epilepsies are not available; however, we present the trends in the section 3.2 according to the remarks of the reviewer.

For section 3.4, it would be interesting to define out of the groups in table 2; which disease groups are most treatable or not. The discussion of this would relate to line 238. Perhaps, the authors can use this information to say where more research needs to be done among the 'metabolic epilepsy' field.

Thank you for the very relevant suggestion. We have extensively described IMD groups with the largest number of treatable metabolic epilepsies, included these groups into the table 3 and added an explanation into the recommended line of Discussion, according to the suggestion.

Some specific minor concerns of mine below :

Some of the description of the results are very hard to read and understand. For example: line 70-91 are very dense and hard to compare - perhaps a table to summarise the proportion and percentage from these various studies?

Table was provided according to the suggestion.

Description of many of the results are not ordered from the highest to the lowest; making appreciation of the trend in the data a bit harder. I suggest the following lines to be revised to be in order from highest to lowest: lines 117-129, lines 158-167.

Revised according to the suggestion.

Figure 1 and Figure 1 should be reordered to make the charts in order from highest to lowest; similar to point 2. Also for Figure 1; perhaps a numerical percentage at the end of each chart would make interpretation of each bar chart easier?

Revised according to the suggestion.

Line 148 and 150 - percentage should also be there next to the number?

Line 51 - should be systemic instead of systematic?

Line 104 - 'supplemented by literature searches as required'?

Corrected.

Line 193 - author should clarify if 'do not present with any changes in routine laboratory or metabolic tests' mean there is no diagnostic markers or that the disease needs molecular testing to diagnose.

Line 220 - 'IMDs such as congenital disorders ....'

Corrected.

Reviewer 2 Report

The manuscript is well written and organized. Cheers to Authors!

I have one minor comment

1. Please include the specific antiepileptic drugs (if reported)/pharmacological treatment used for different metabolic epilepsies in Table 3. 

Author Response

Reviewer 2:

The manuscript is well written and organized. Cheers to Authors!

I have one minor comment

  1. Please include the specific antiepileptic drugs (if reported)/pharmacological treatment used for different metabolic epilepsies in Table 3.

Thank you very much for the opinion and remarks. As we did not provide the names of specific treatments in “Table 4. Treatable metabolic epilepsies”, we could not include specific AEDs into the table itself, but we have now added this information in section 3.4, Treatable Metabolic Epilepsies.